# Open-Ended 3D Metric-Semantic Representation Learning via Semantic-Embedded Gaussian Splatting

## Abstract

This work answers the question of whether it is feasible to create a comprehensive metric-semantic 3D virtual world using everyday devices equipped with multi-view stereo. We propose an open-ended metric-semantic representation learning framework based on 3D Gaussians, which distills open-set semantics from 2D foundation models into a scalable and continuously evolving 3D Gaussian representation, optimized within a SLAM framework. The process is nontrivial. The scalability requirements make direct embedding of semantic information into Gaussians impractical, resulting in excessive memory usage and semantic inconsistencies. In response, we propose to learn semantics by aggregating from a condensed, fixed-sized semantic pool rather than directly embedding high-dimensional raw features, significantly reducing memory requirements compared to the point-wise representation. Additionally, by enforcing pixel-to-pixel and pixel-to-object semantic consistency through contrastive learning and stability-guided optimization, our framework enhances coherence and stability in semantic representations. Extensive experiments demonstrate that our framework presents a precise open-ended metric-semantic field with superior rendering quality and tracking accuracy. Besides, it accurately captures both closed-set object categories and open-set semantics, facilitating various applications, notably fine-grained, unrestricted 3D scene editing. These results mark an initial yet solid step towards efficient and expressive 3D virtual world modeling. Our code will be released.

## 1 Introduction

A 3D virtual world functions as a collective space where user avatars interact seamlessly within a metric-semantic representation of 3D environments, encompassing both appearance and semantics (Gupta et al., 2009; Dionisio et al., 2013). Recent technological strides, seen in platforms like VisionPro (Apple, 2023) and Metaverse (Meta, 2024), signify a transition towards richer content, vast expanses, and the ambitious goal of encompassing the entire Earth within this digital realm (Li et al., 2023a; Huang et al., 2023; Puig et al., 2023; Wang et al., 2024; Cai et al., 2024). This progression raises a fundamental query: *is it possible to develop an open-ended metric-semantic 3D virtual world using everyday devices equipped with multi-view stereo?*

Such a system should not only capture the spatial layout but also incorporate semantic information crucial for meaningful interactions. It must possess three essential characteristics: *i)* scalability to adapt limitlessly to evolving environments; *ii)* open-ended semantic framework capable of accommodating new concepts and free-form queries; *iii)* efficiency in rendering speed and memory usage to ensure portability as scene complexity increases. While recent breakthroughs like Neural Radiance Fields (NeRFs) (Mildenhall et al., 2020) and Diffusion models (Nichol & Dhariwal, 2021; Rombach et al., 2022) have enabled photorealistic representations of intricate 3D scenes, yet their slow training and inference speeds impede their practicality (Barron et al., 2021; 2022; Chen et al., 2023; Sun et al., 2022; Müller et al., 2022), particularly in expanding applications. On the other hand, recent progress such as 3D Gaussian Splatting (3DGS) (Kerbl et al., 2023) has notably enhanced training and rendering speeds, holding promise for real-time novel view synthesis. Nonetheless, these advancements focus primarily on visual fidelity, still overlooking the intrinsic semantics within 3D

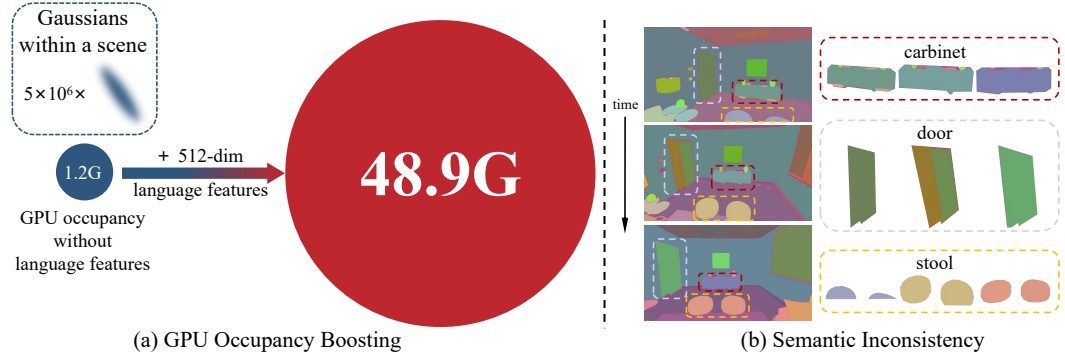

(a) GPU Occupancy Boosting

(b) Semantic Inconsistency

Figure 1: Two inherent challenges arise in the open-ended 3D metric-semantic representation learning. (a) Assigning a semantic feature to each Gaussian is inefficient and consumes excessive GPU memory. For instance, allocating 512-dimensional features per Gaussian in a scene increases GPU occupancy by $\sim$40$\times$. (b) Inconsistencies in open-set semantic features across frames hinder the learning of a coherent semantic field.

environments (Yu et al., 2024; Sun et al., 2024a; Li et al., 2024a). The challenge of constructing a scalable and semantically rich 3D scene representation remains a significant, unresolved problem.

This work aims to bridge this gap by presenting an open-ended 3D metric-semantic representation learning framework based on 3DGS. The core idea is to distill open-set semantics from 2D foundation models like CLIP (Radford et al., 2021) or SAM (Kirillov et al., 2023) into a scalable and continuously evolving 3D Gaussian representation, optimized within a simultaneous localization and mapping (SLAM) framework. One straightforward approach involves embedding feature dimensions to each 3D Gaussian, alongside color information, to represent additional semantics. However, this process encounters two inherent challenges as environments evolve gradually. **First**, memory and computation consumption can become prohibitively large, significantly reducing optimization and rendering efficiency with respective to the number of Gaussian points and the dimension of the learned feature. Existing solutions often restrict feature dimensions to a smaller scale (Qin et al., 2024), *i.e.*, 3 *vs.* 512 (the original feature dimension of CLIP), to mitigate this issue, albeit at the cost of semantic expressiveness. **Second**, semantic inconsistency across viewpoints. As depicted in Fig. 1, due to the 2D nature of off-the-shelf foundation models, ensuring semantic coherence for identical objects across continuous view renderings cannot be guaranteed, thereby impeding effective learning of semantic fields.

Focusing on the aforementioned two intrinsic challenges, we first integrate the Semantic Feature Aggregation mechanism into the framework. Instead of directly expanding the semantic feature channel, this module associates each Gaussian with a low-dimensional key vector to access point-specific semantics from a fixed-size learnable semantic feature pool. This pool holds a condensed representation of open-set semantics across the entire scene, significantly smaller in scale compared to the number of Gaussian points, *i.e.*, 200 *vs.* $5 \times 10^{6}$. This mechanism not only mitigates the substantial memory requirements but also diminishes the redundancy of local semantic features. Given that local Gaussians representing the same object should exhibit similar semantics, this approach promotes more effective semantic field learning. Next, to improve semantic consistency during optimization, the approach is intuitive: ensuring that the semantic representation of any pixel in a 2D image aligns with neighboring pixels from the same object (*intra*-frame) and pixels at the same 3D locations aligned with camera poses from previous frames (*inter*-frame). We employ contrastive learning to enforce these correspondences and introduce the Intra-Inter Semantic Consistency Objective during semantic field learning. Finally, to further address semantic ambiguity, which arises when different semantic labels are attributed to the same object, hindering field learning, we introduce Semantic Stability Guidance. By measuring the inter-frame pixel-to-object semantic consistency through cosine similarity, we leverage this metric to adjust the learning signal. Signals from inconsistent regions are reduced, while those from consistent regions are amplified, enhancing the overall coherence and stability of semantic representations within the framework.

For examination, we compare our framework with both NeRF-based (Yang et al., 2022; Zhu et al., 2022; Johari et al., 2023; Wang et al., 2023; Sandström et al., 2023) and 3DGS-based (Yan et al., 2024; Keetha et al., 2024; Li et al., 2024c) SLAM methods on the Replica dataset (Straub et al., 2019) in §3.2. Experimental results demonstrate that our framework establishes a precise open-

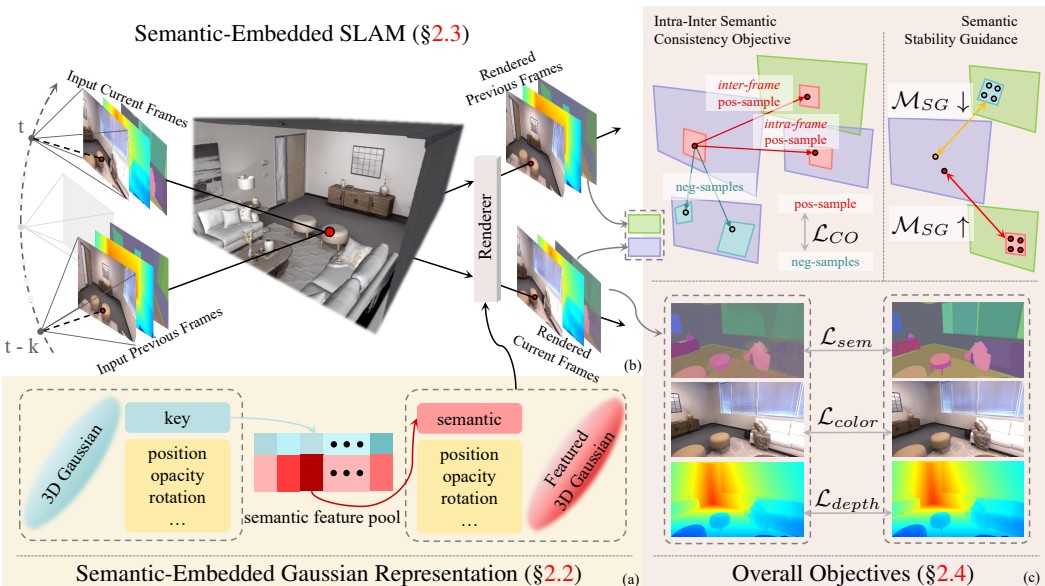

Figure 2: **Framework**. (a) Each Gaussian is linked to a low-dimensional key vector that retrieves point-specific semantics from a fixed-size learnable feature pool. (b) The SLAM framework processes RGB-D frames alongside a semantic feature map generated by a pre-trained model. (c) To enhance semantic consistency and address semantic ambiguity during optimization, we introduce the Intra-Inter Semantic Consistency Objective and Semantic Stability Guidance.

ended metric-semantic field, exhibiting superior rendering quality and tracking accuracy. It showcases the ability to accurately capture not only closed-set object categories (§3.3) but also open-set semantics, facilitating a wide range of applications such as 3D scene editing (§3.4). This functionality empowers fine-grained and unrestricted semantic-based manipulation of target objects without affecting the surrounding environment, a capability rarely reported in previous studies.

## 2 METHODOLOGY

### 2.1 FRAMEWORK OVERVIEW

Given an RGB-D stream, our framework aims to reconstruct a scalable Gaussian field enriched with open-set semantic information. Since assigning a high-dimensional semantic feature to each Gaussian is both computationally expensive and spatially redundant, we introduce the Semantic Feature Aggregation, allocating semantic features only to Gaussians involved in rendering and optimization. To further address inconsistencies of the open-set semantic features generated across frames, we introduce the Intra-Inter Semantic Consistency Objective technique. Additionally, to reduce the impact of inaccurate new-income semantic features to the reconstructed semantic field, we develop a Semantic Stability Guidance. An overview of our framework is presented in Fig. 2.

### 2.2 SEMANTIC-EMBEDDED GAUSSIAN REPRESENTATION

3DGS (Kerbl et al., 2023) represents an explicit 3D scene with a set of Gaussians. In our work, we simplify the Gaussians to be isotropic for more efficient scene representation:

$$g(\boldsymbol{x}) = \sigma \exp\left(-\frac{\|\boldsymbol{x} - \boldsymbol{\mu}\|^2}{2r^2}\right), \tag{1}$$

where $\mu \in \mathbb{R}^3$, $r$ and $\sigma$ indicates the position, radius and opacity of the Gaussian in 3D space.

**Color and Depth Rendering.** Given a set of 3D Gaussians and a camera pose, the first step is to sort all Gaussians. The influence of all Gaussians on a certain pixel can be integrated by performing front-to-back volume rendering. Images can be generated by applying alpha-compositing to the

splatted 2D projection of each Gaussian in a sequential manner within the pixel space. The center $\mu$ and the radius $r$ of each Gaussian can be written with the depth of the $i^{th}$ Gaussian $d$ as:

$$\boldsymbol{\mu}^{2D} = K_c \frac{E_t \boldsymbol{\mu}}{d}, r^{2D} = \frac{fr}{d}, \text{where } d = (E_t \boldsymbol{\mu})_z, \tag{2}$$

where $K_c$ represents the camera intrinsic matrix, $E_t$ represents the extrinsic matrix capturing the rotation and translation of the camera at frame $t$, and $f$ is the focal length. Thus, the rendered color of a particular pixel $\boldsymbol{p}$ can be computed with $N$ splatted 2D Gaussians as:

$$C(\boldsymbol{p}) = \sum_{i=1}^{N} c_i g_i(\boldsymbol{p}) \prod_{j=1}^{i-1}(1 - g_j(\boldsymbol{p})), \tag{3}$$

where $c_i$ denotes the RGB color of each Gaussian and $f_i(\boldsymbol{p})$ is computed via Eq. 1. Similarly, we differentiably render depth with:

$$D(\boldsymbol{p}) = \sum_{i=1}^{N} d_i g_i(\boldsymbol{p}) \prod_{j=1}^{i-1}(1 - g_j(\boldsymbol{p})), \tag{4}$$

where $d_i$ represents the depth of each Gaussian.

**Semantic Feature Aggregation.** We enhance each Gaussian with an additional semantic feature. Previous NeRF-based methods (Kerr et al., 2023) often directly integrate high-dimensional semantic features into the scene. However, assigning such features to every Gaussian greatly increases memory demands, and many Gaussians are inactive during rendering, which reduces efficiency. Additionally, unlike color maps, objects of the same category share identical semantics, making redundant features unnecessary. To address this, we propose using a semantic feature pool to reduce memory and assign features efficiently to active Gaussians.

Specifically, we assign a key $\hat{k}$ to each Gaussian, projecting keys of active Gaussians into high-dimensional features. To facilitate this, we develop a learnable key pool $\mathcal{K} = \{k_l | l = 1, 2, \cdots, M\}$ and a learnable high-dimensional semantic feature pool $\mathcal{F} = \{f_l | l = 1, 2, \cdots, M\}$, both of size $M$. We calculate the similarity between each Gaussian's key $\hat{k}$ and the keys in the key pool, aggregating the semantic feature $\hat{f}$ from the most similar key:

$$\hat{f} = \sum_{l=1}^{M} \hat{f}_l \cdot \text{softmax}(\text{simVec}(\hat{k}, \mathcal{K})), \tag{5}$$

where $\text{simVec}(\hat{k}, \mathcal{K})$ represents the similarity scores. This approach minimizes memory overhead while ensuring that the Gaussians retain high-dimensional semantic features.

**Semantic Feature Rendering.** Within the rendering phase, our method is capable of rendering a 2D semantic feature map from the 3D scene, following the rendering process of the color map:

$$F(\boldsymbol{p}) = \sum_{i=1}^{N} \hat{f}_i g_i(\boldsymbol{p}) \prod_{j=1}^{i-1}(1 - g_j(\boldsymbol{p})), \tag{6}$$

where $\hat{f}_i$ represents the semantic feature of each Gaussian.

## 2.3 SEMANTIC-EMBEDDED SLAM FRAMEWORK

Our SLAM framework begins by processing RGB images to generate semantic feature frames with pre-trained models (*supp.* B.1). Given a Gaussian field constructed from frames 1 to $t$, along with new RGB, depth, semantic feature frames at $t + 1$, the framework performs tracking and mapping.

**Tracking.** Camera tracking determines the current camera position using incoming data to estimate relative motion. During tracking, only camera parameters are optimized, with Gaussians parameters fixed. When initializing with the first frame, the camera tracking stage is bypassed. For each subsequent timestep, the camera pose is estimated by forward-projecting pose parameters from the camera center into quaternion space. Consequently, the camera parameters are initialized as:

$$E_{t+1} = E_t + (E_t - E_{t-1}). \tag{7}$$

**Mapping.** The mapping process generates an open-ended metric-semantic spatial representation of the scene. Using the camera's tracked position and depth frames, it refines the map by adding Gaussians to underdeveloped areas. Unlike in the tracking phase, the poses remain fixed during mapping, while Gaussian parameters are updated. We optimize these parameters through differentiable rendering of RGB, depth, and semantic feature frames using gradient-based techniques. Large or low-opacity Gaussians are removed, as outlined in (Kerbl et al., 2023).

## 2.4 OVERALL OBJECTIVES

Given a reconstructed field from frames 1 to $t$, along with rendered RGB, depth, semantic feature frames $C$, $D$, $F$ and new RGB, depth, semantic feature frames $C_{GT}$, $D_{GT}$, $F_{GT}$ at $t + 1$, our framework optimizes using tracking and mapping losses. For the mapping loss, we include two key components: the Intra-Inter Semantic Consistency Objective and Semantic Stability Guidance, ensuring accurate semantic field optimization.

**Tracking Loss.** This process relies on the differentiable rendering of RGB, depth, and semantic maps via Eqs. 3, 4 and 6. For each pixel $\boldsymbol{p}$, let $C_{GT}(\boldsymbol{p})$, $D_{GT}(\boldsymbol{p})$ and $F_{GT}(\boldsymbol{p})$ represent the ground truth RGB, depth, and semantic feature, respectively. We optimize camera parameters with:

$$\mathcal{L}_{tracking} = \sum_{\boldsymbol{p}} (\lambda_C^T \mathcal{L}_{color}^T + \lambda_D^T \mathcal{L}_{depth}^T + \lambda_F^T \mathcal{L}_{sem}^T), \tag{8}$$

where

$$\mathcal{L}_{color}^T = \|C(\boldsymbol{p}) - C_{GT}(\boldsymbol{p})\|, \mathcal{L}_{depth}^T = \|D(\boldsymbol{p}) - D_{GT}(\boldsymbol{p})\|, \mathcal{L}_{sem}^T = \|F(\boldsymbol{p}) - F_{GT}(\boldsymbol{p})\|. \tag{9}$$

We utilize the loss only on pixels from well-reconstructed parts of the map.

**Intra-Inter Semantic Consistency Objective.** The semantic features derived from the open-set approach are better suited for 2D images than for 3D scenes. While this allows for reconstructing a semantic field from RGB-D inputs, it can also lead to inconsistencies of semantic features generated between frames—particularly at objects from the edges of images. Such inconsistencies pose challenges for accurately reconstructing the open-set semantic field. Since the semantic features of the same object should remain consistent, we address this issue with a contrastive learning approach.

For a given pixel $\boldsymbol{p}$ in frame $F$, it should have the same semantic feature as other pixels belonging to the same object. Based on this principle, we select a pixel from *intra-frame* that shares the same object and treat it as a positive sample. Additionally, $\boldsymbol{p}$ should also maintain the same semantic feature as the corresponding pixel in previous frames which has the same projection position in 3D space. To accomplish this, we project $\boldsymbol{p}$ into 3D space using the depth map and then reproject it back onto the previous frame, to find the corresponding pixel in the *inter-frame*, which serves as a second positive sample. Remaining pixels from $F$ are added to the negative sample pool.

Following this approach, we selective a positive sample $\boldsymbol{p}^+$ mentioned above and a negative sample pool $\mathcal{N}^-$ containing negative samples $\boldsymbol{p}_i^-$, to compute the semantic consistency objective:

$$\mathcal{L}_{CO} = -\log \frac{\exp(\boldsymbol{p} \cdot \boldsymbol{p}^+)}{\sum_{\boldsymbol{p}_i^- \in \mathcal{N}^-} \exp(\boldsymbol{p} \cdot \boldsymbol{p}_i^-)}. \tag{10}$$

**Semantic Stability Guidance.** We continue to address semantic ambiguity between frames. When given a Gaussian field built from frames 1 to $t$ and inputting $F_{GT}$ at $t + 1$, inconsistencies in the same object's features can negatively impact the existing semantic field. To mitigate this, we propose Semantic Stability Guidance to reduce the influence of inaccurate features.

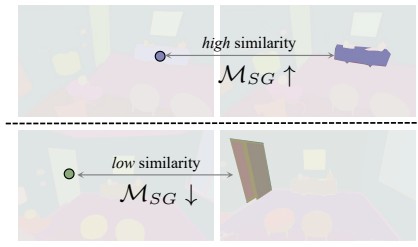

Figure 3: Semantic Stability Guidance.

For a pixel $\boldsymbol{p}$ in frame $F$, we use the same projection rules from the Intra-Inter Semantic Consistency Objective to locate the corresponding pixel in the previous semantic frame, identifying its segmentation region. As in Fig. 3, We calculate the cosine similarity between $F(\boldsymbol{p})$ and the average feature of pixels in this region, using this similarity as the Semantic Stability Guidance $\mathcal{M}_{SG}(\boldsymbol{p})$. In areas of high similarity, the loss remains larger for normal optimization, while in low-similarity areas, it is reduced to limit negative impacts on correct reconstructions.

**Overall Mapping Loss.** Different from Eq. 8, we optimizes a mapping loss that all pixels are calculated. Specifically, we calculate the color loss with an additional SSIM (Wang et al., 2004) loss:

$$\mathcal{L}_{color}^M = \lambda_1^c \|C(\boldsymbol{p}) - C_{GT}(\boldsymbol{p})\| + \lambda_2^c (1 - \text{SSIM}(C(\boldsymbol{p}), C_{GT}(\boldsymbol{p}))). \tag{11}$$

We get the semantic feature loss with Eq. 10:

$$\mathcal{L}_{sem}^M = \lambda_1^s \|F(\boldsymbol{p}) - F_{GT}(\boldsymbol{p})\| \cdot \mathcal{M}_{SG}(\boldsymbol{p}) + \lambda_2^s \mathcal{L}_{CO}. \tag{12}$$

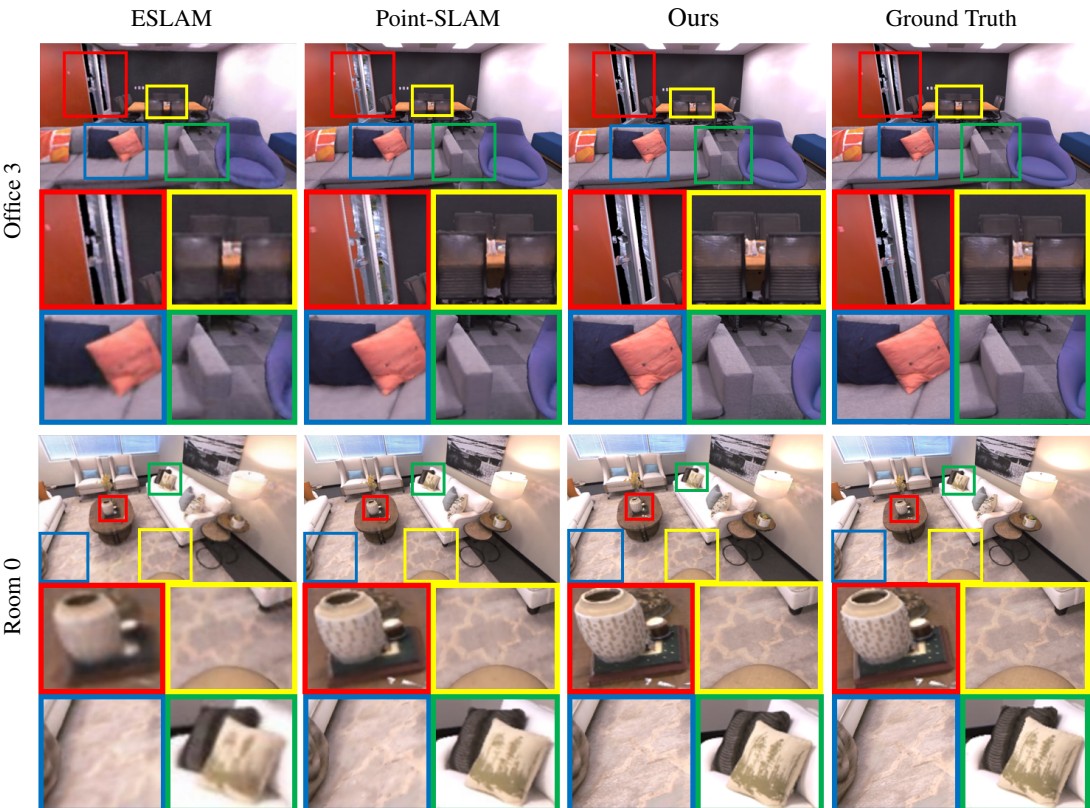

ESLAM  Point-SLAM  Ours  Ground Truth

Office 3

Room 0

Figure 4: **Qualitative Comparisons** (§3.4) on Replica (Straub et al., 2019).

While depth loss can be calculated with

$$\mathcal{L}_{depth}^M = \| D(\boldsymbol{p}) - D_{GT}(\boldsymbol{p}) \|, \tag{13}$$

combining Eqs. 11, 12, 13, the final mapping loss is:

$$\mathcal{L}_{mapping} = \lambda_C^M \mathcal{L}_{color}^M + \lambda_D^M \mathcal{L}_{depth}^M + \lambda_F^M \mathcal{L}_{sem}^M. \tag{14}$$

## 3 EXPERIMENT

### 3.1 EXPERIMENTAL SETUP

**Datasets.** Experiments are carried out on eight scenes of Replica (Straub et al., 2019). Results on real-world data TUM (Sturm et al., 2012) and ScanNet (Dai et al., 2017) are delivered in *supp.* C.

**Metrics.** We adopt a standardized set of metrics to evaluate both camera pose estimation and rendering performance. Camera pose tracking is assessed by the average absolute trajectory error (ATE RMSE) (Sturm et al., 2012). RGB rendering performance is measured using PSNR, SSIM (Wang et al., 2004), and LPIPS (Zhang et al., 2018), while reconstruction performance is evaluated using Depth L1. For semantic segmentation, we employ the mean Intersection over Union (mIoU) as our evaluation metric. In all tables, the best results are marked as **first** and second.

**Baselines.** We compare our approach to latest NeRF-based and 3DGS-based methods. Among NeRF-based methods, our competitors include Vox-Fusion (Yang et al., 2022), NICE-SLAM (Zhu et al., 2022), ESLAM (Johari et al., 2023), Co-SLAM (Wang et al., 2023), and Point-SLAM (Sandström et al., 2023). Within the 3DGS framework, we compare our method with GS-SLAM (Yan et al., 2024), SplaTAM (Keetha et al., 2024), and SGS-SLAM (Li et al., 2024c). Additionally, we compare our mIoU results with several semantic SLAM approaches, including NIDS-SLAM (Haghighi et al., 2023), DNS-SLAM (Li et al., 2023b), SNI-SLAM (Zhu et al., 2024b), SGS-SLAM (Li et al., 2024c). Notably, existing semantic SLAM frameworks focus solely on closed-set segmentation, whereas our approach learns an open-set feature field. This enables us to handle not

Table 1: **Quantitative Comparisons on Rendering Performance** (§3.2) with baselines on Replica (Straub et al., 2019).

| Methods | Metrics | Avg. | R0 | R1 | R2 | Of0 | Of1 | Of2 | Of3 | Of4 |
|---|---|---|---|---|---|---|---|---|---|---|
| Vox-Fusion [ISMAR22] (Yang et al., 2022) | PSNR↑ | 24.41 | 22.39 | 22.36 | 23.92 | 27.29 | 29.83 | 20.33 | 23.47 | 25.21 |
| | SSIM↑ | 0.801 | 0.683 | 0.751 | 0.798 | 0.857 | 0.876 | 0.794 | 0.803 | 0.847 |
| | LPIPS↓ | 0.236 | 0.303 | 0.269 | 0.234 | 0.184 | 0.184 | 0.243 | 0.213 | 0.199 |
| NICE-SLAM [CVPR22] (Zhu et al., 2022) | PSNR↑ | 24.42 | 22.12 | 22.47 | 24.52 | 29.07 | 30.34 | 19.66 | 22.23 | 24.94 |
| | SSIM↑ | 0.809 | 0.689 | 0.757 | 0.814 | 0.874 | 0.886 | 0.797 | 0.801 | 0.856 |
| | LPIPS↓ | 0.233 | 0.330 | 0.271 | 0.208 | 0.229 | 0.181 | 0.235 | 0.209 | 0.198 |
| ESLAM [CVPR23] (Johari et al., 2023) | PSNR↑ | 29.08 | 25.32 | 27.77 | 29.08 | 33.71 | 30.20 | 28.09 | 28.77 | 29.71 |
| | SSIM↑ | 0.929 | 0.875 | 0.902 | 0.932 | 0.960 | 0.923 | 0.943 | 0.948 | 0.945 |
| | LPIPS↓ | 0.336 | 0.313 | 0.298 | 0.248 | 0.184 | 0.228 | 0.241 | 0.196 | 0.204 |
| Co-SLAM [CVPR23] (Wang et al., 2023) | PSNR↑ | 30.24 | 27.27 | 28.45 | 29.06 | 34.14 | 34.87 | 28.43 | 28.76 | 30.91 |
| | SSIM↑ | 0.939 | 0.910 | 0.909 | 0.932 | 0.961 | 0.969 | 0.938 | 0.941 | 0.955 |
| | LPIPS↓ | 0.252 | 0.324 | 0.294 | 0.266 | 0.209 | 0.196 | 0.258 | 0.229 | 0.236 |
| Point-SLAM [ICCV23] (Sandström et al., 2023) | PSNR↑ | 35.17 | 32.40 | 34.08 | **35.50** | 38.26 | 39.16 | **33.99** | **33.48** | 33.49 |
| | SSIM↑ | 0.975 | 0.974 | 0.977 | 0.982 | 0.983 | 0.986 | 0.960 | 0.960 | 0.979 |
| | LPIPS↓ | 0.124 | 0.113 | 0.116 | 0.111 | 0.100 | 0.118 | 0.156 | 0.132 | 0.142 |
| GS-SLAM [CVPR24] (Yan et al., 2024) | PSNR↑ | 34.27 | 31.56 | 32.86 | 32.59 | 38.70 | **41.17** | 32.36 | 32.03 | 32.92 |
| | SSIM↑ | 0.975 | 0.968 | 0.973 | 0.971 | 0.986 | **0.993** | 0.978 | 0.970 | 0.968 |
| | LPIPS↓ | 0.082 | 0.094 | 0.075 | 0.093 | 0.050 | **0.033** | 0.094 | 0.110 | 0.112 |
| SplaTAM [CVPR24] (Keetha et al., 2024) | PSNR↑ | 34.11 | 32.86 | 33.89 | 35.25 | 38.26 | 39.17 | 31.97 | 29.70 | 31.81 |
| | SSIM↑ | 0.970 | **0.980** | 0.970 | 0.980 | 0.980 | 0.980 | 0.970 | 0.950 | 0.950 |
| | LPIPS↓ | 0.100 | 0.070 | 0.100 | 0.080 | 0.090 | 0.090 | 0.100 | 0.120 | 0.150 |
| SGS-SLAM [ECCV24] (Li et al., 2024c) | PSNR↑ | 34.66 | 32.50 | 34.25 | 35.10 | 38.54 | 39.20 | 32.90 | 32.05 | 32.75 |
| | SSIM↑ | 0.973 | 0.976 | 0.978 | 0.981 | 0.984 | 0.980 | 0.967 | 0.966 | 0.949 |
| | LPIPS↓ | 0.096 | 0.070 | 0.094 | 0.070 | 0.086 | 0.087 | 0.101 | 0.115 | 0.148 |
| **Ours** | PSNR↑ | **35.80** | **33.16** | **34.90** | 35.43 | **40.20** | 40.61 | **33.65** | 32.59 | **35.89** |
| | SSIM↑ | **0.984** | 0.979 | **0.986** | **0.985** | **0.989** | 0.989 | **0.981** | **0.977** | **0.984** |
| | LPIPS↓ | **0.060** | **0.061** | **0.043** | **0.069** | **0.044** | 0.046 | **0.074** | **0.063** | **0.076** |

just closed-set segmentation, but also tasks like 3D scene editing (§3.4). For fair comparison, we compute mIoU using an additional segmentation head (*supp.* B.2) on top of the learned semantic features, supervised by ground-truth labels.

**Implementation Details.** Our experiments run on a server with a single NVIDIA GeForce RTX 3090 GPU. The dimension of key $\hat{k}$, the dimension of semantic feature $\hat{f}$, and the size $M$ of pools are set to 3, 16, and 200, respectively. During the SLAM process, tracking is performed for each frame, while mapping is conducted only at selected mapping frames (*supp.* B.2). The optimization involves ten parameters, $\lambda_C^T = \lambda_C^M = 0.5$, $\lambda_D^T = \lambda_D^M = 1.0$, $\lambda_F^T = \lambda_F^M = 0.05$, $\lambda_1^c = 0.8$, $\lambda_2^c = 0.2$, $\lambda_1^s = 0.999$, and $\lambda_2^s = 0.001$. During tracking, only the camera parameters are optimized, with learning rates of 2e-3 for translations of camera poses, and 4e-4 for unnormalized rotations. During mapping, only Gaussian parameters are optimized, with learning rates of 9e-5 for 3D positions, 2.5e-3 for colors, 1e-1 for semantic-embedded parameter, 1e-2 for semantic feature pool, 1e-3 for key pool, 8e-4 for rotations, 4e-2 for opacities, and 8e-4 for scales. Iterations of tracking and mapping are 40 and 60 for Replica (Straub et al., 2019). The code will be released.

## 3.2 QUANTITATIVE RESULTS ON TRACKING AND RENDERING

**Comparison of Camera Pose Estimation.** Table 2 illustrates the improvements achieved by our method over previous NeRF-based and 3DGS-based approaches on Replica. The 3DGS field is capable of representing scenes with greater accuracy than the NeRF field, allowing 3DGS-based methods to estimate camera trajectories more precisely than NeRF-based methods. Furthermore, the inclusion of semantic information provides the system with more details about the scene, enabling even more accurate camera pose estimation. This results

Table 2: **Quantitative Comparisons on Camera Pose Estimation** (§3.2) with baselines on Replica (Straub et al., 2019) (ATE RMSE↓ [cm]).

| Methods | Avg. | R0 | R1 | R2 | Of0 | Of1 | Of2 | Of3 | Of4 |
|---|---|---|---|---|---|---|---|---|---|
| Vox-Fusion (Yang et al., 2022) | 3.09 | 1.37 | 4.70 | 1.47 | 8.48 | 2.04 | 2.58 | 1.11 | 2.94 |
| NICE-SLAM (Zhu et al., 2022) | 1.06 | 0.97 | 1.31 | 1.07 | 0.88 | 1.00 | 1.06 | 1.10 | 1.13 |
| ESLAM (Johari et al., 2023) | 0.63 | 0.71 | 0.70 | 0.52 | 0.57 | 0.55 | 0.58 | 0.72 | 0.63 |
| Co-SLAM (Wang et al., 2023) | 0.86 | 0.65 | 1.13 | 1.43 | 0.55 | 0.50 | 0.46 | 1.40 | 0.77 |
| Point-SLAM (Sandström et al., 2023) | 0.52 | 0.61 | 0.41 | 0.37 | 0.38 | 0.48 | 0.54 | 0.69 | 0.72 |
| GS-SLAM (Yan et al., 2024) | 0.50 | 0.48 | 0.53 | 0.33 | 0.52 | 0.41 | 0.59 | 0.46 | 0.70 |
| SplaTAM (Keetha et al., 2024) | 0.36 | 0.31 | 0.40 | 0.29 | 0.47 | 0.27 | 0.29 | 0.32 | 0.55 |
| **Ours** | **0.30** | **0.24** | **0.39** | **0.28** | **0.29** | **0.18** | **0.25** | **0.30** | **0.46** |

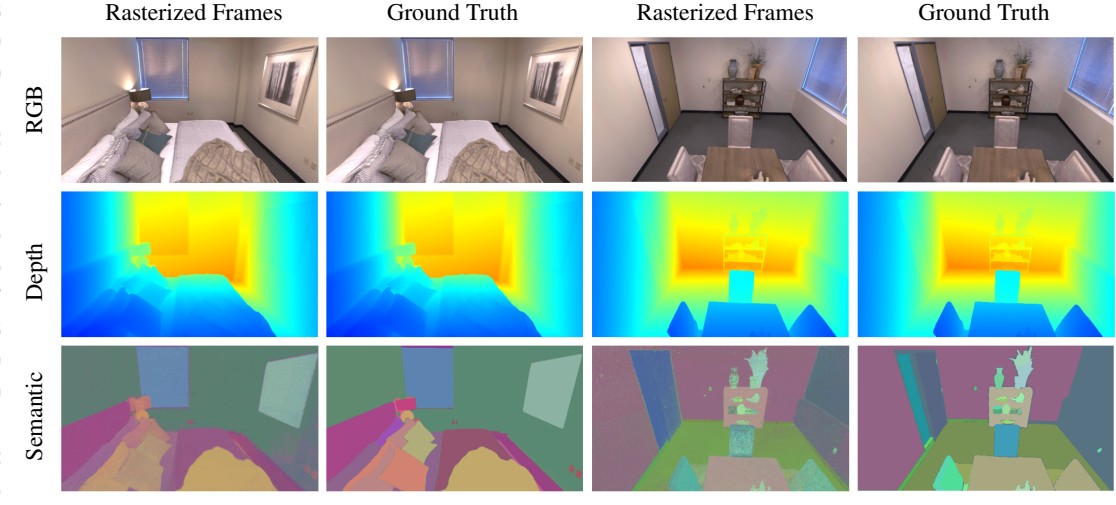

Figure 5: **Qualitative Results** (§3.4) on Replica (Straub et al., 2019).

in performance improvements over existing 3DGS-based methods without semantic input, such as GS-SLAM (Yan et al., 2024) and SplaTAM (Keetha et al., 2024).

**Quantitative Analysis of Rendering.** Table 1 demonstrates that our method consistently outperforms a range of NeRF-based approaches, including the state-of-the-art Point-SLAM (Sandström et al., 2023). Although our PSNR does not surpass Point-SLAM in all scenes, we achieve significantly higher SSIM and LPIPS scores. Compared to GS-SLAM (Yan et al., 2024) and SplaTAM (Keetha et al., 2024), the inclusion of semantic input allows our method to better capture scene features such as object shapes, further enhancing reconstruction quality. Additionally, the contrastive learning technique we have developed introduces effective object-level discrimination, providing richer information to the system. As a result, we also outperform SGS-SLAM (Li et al., 2024c).

**Quantitative Results on Depth L1 Error.** To evaluate the geometric reconstruction accuracy, we also assess the Depth L1 error, as

Table 3: **Quantitative Comparisons on Reconstruction Performance** (§3.2) on Replica (Straub et al., 2019) (Depth L1↓ [cm]).

| Methods | Avg. | R0 | R1 | R2 | Of0 | Of1 | Of2 | Of3 | Of4 |
|---|---|---|---|---|---|---|---|---|---|
| Vox-Fusion (Yang et al., 2022) | 2.46 | 1.09 | 1.90 | 2.21 | 2.32 | 3.40 | 4.19 | 2.96 | 1.61 |
| NICE-SLAM (Zhu et al., 2022) | 2.97 | 1.81 | 1.44 | 2.04 | 1.39 | 1.76 | 8.33 | 4.99 | 2.01 |
| ESLAM (Johari et al., 2023) | 1.18 | 0.97 | 1.07 | 1.28 | 0.86 | 1.26 | 1.71 | 1.43 | 1.06 |
| Point-SLAM (Sandström et al., 2023) | 0.44 | 0.53 | **0.22** | 0.46 | 0.30 | 0.57 | 0.49 | 0.51 | **0.46** |
| GS-SLAM (Yan et al., 2024) | 1.16 | 1.31 | 0.82 | 1.26 | 0.81 | 0.96 | 1.41 | 1.53 | 1.08 |
| **Ours** | **0.43** | **0.48** | 0.42 | **0.43** | **0.28** | **0.39** | **0.43** | **0.47** | 0.54 |

shown in Table 3. Our approach outperforms other NeRF-based methods and achieves comparable results to the state-of-the-art Point-SLAM (Sandström et al., 2023). Furthermore, we surpass the 3DGS-based GS-SLAM (Yan et al., 2024), highlighting the robustness of our method.

### 3.3 Quantitative Results on Semantic Reconstruction

Table 4 provides a quantitative comparison between our method and several semantic SLAM frameworks across four scenes from Replica (Straub et al., 2019). The results are obtained with additional segmentation

Table 4: **Quantitative Comparisons on Semantic Reconstruction Accuracy** (§3.3) on Replica (Straub et al., 2019) (mIoU ↑ [%]).

| Methods | Avg. | R0 | R1 | R2 | Of0 |
|---|---|---|---|---|---|
| NIDS-SLAM (Haghighi et al., 2023) | 82.37 | 82.45 | 84.08 | 76.99 | 85.94 |
| DNS-SLAM (Li et al., 2023b) | 84.77 | 88.32 | 84.90 | 81.20 | 84.66 |
| SNI-SLAM (Zhu et al., 2024b) | 87.41 | 88.42 | 87.43 | 86.16 | 87.63 |
| SGS-SLAM (Li et al., 2024c) | 92.72 | 92.95 | 92.91 | 92.10 | 92.90 |
| **Ours** | **94.38** | **95.14** | **94.16** | **93.89** | **94.32** |

head on top of learned semantic feature (*supp.* B.2). To ensure a fair comparison, we use ground-truth labels to supervise such head following convensions. As illustrated in Table 4, our method surpasses existing NeRF-based approaches. Furthermore, the use of Intra-Inter Semantic Consistency Objective and Semantic Stability Guidance enhances our ability to delineate object

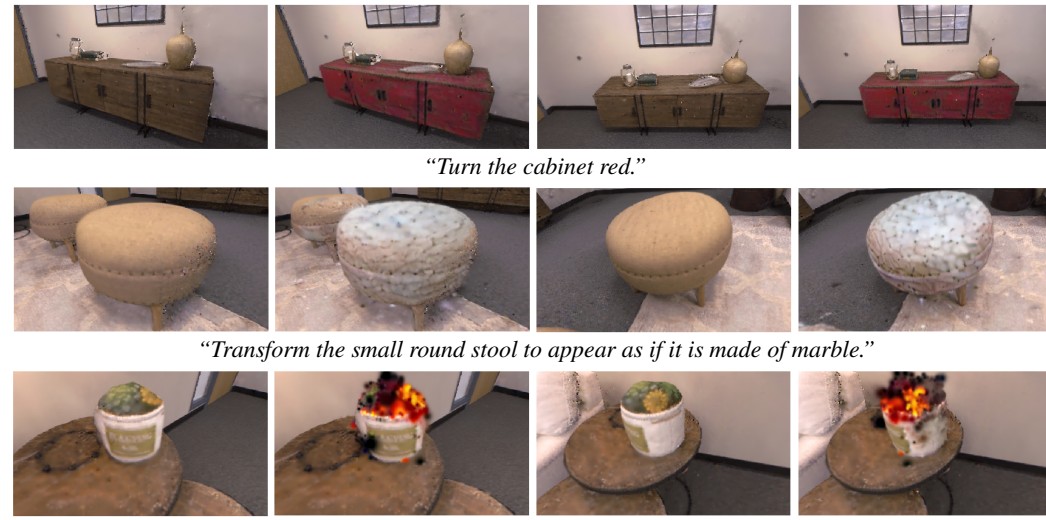

*"Turn the cabinet red."*

*"Transform the small round stool to appear as if it is made of marble."*

*"Make the flower burn."*

Figure 6: **Qualitative Results of 3D Editing** (§3.4) on Replica (Straub et al., 2019).

boundaries more effectively and track objects consistently across frames. Such improvement enables us to outperform SGS-SLAM (Li et al., 2024c), resulting in more precise semantic representations.

### 3.4 QUALITATIVE RESULTS

**Qualitative Comparisons on Scene Reconstruction.** The visual comparisons on Replica (Straub et al., 2019), as depicted in Fig. 4, demonstrate the superior performance of our method. Compared to ESLAM (Johari et al., 2023), our approach produces noticeably clearer reconstructions. When benchmarked against Point-SLAM (Sandström et al., 2023), our method excels at capturing finer details, such as patterns, wrinkles, plush textures, and even lighting variations. For more qualitative comparisons on TUM (Sturm et al., 2012) and ScanNet (Dai et al., 2017), please refer to *supp.* C.2.

**More Qualitative Results.** More visual results on Replica (Straub et al., 2019) are presented in Fig. 5. As seen, we are able to reconstruct high-quality color maps with high geometric reconstruction accuracy. We are also able to reconstruct a semantic feature field with clear edges, which is beneficial for downstream tasks such as 3D scene editing. For more comprehensive qualitative results on Replica (Straub et al., 2019), as well as results on TUM (Sturm et al., 2012) and ScanNet (Dai et al., 2017), please refer to *supp.* C.2.

**Qualitative Results on 3D Editing.** We further explore the impact of our 3D editing capabilities on the Replica dataset (Straub et al., 2019). To visually demonstrate the effectiveness and accuracy of our editing mechanism, we present a series of edited scenes that showcase the ability to edit objects within any given environment. The positive outcomes indirectly affirm our method's ability to construct a coherent semantic field throughout the SLAM process.

As illustrated in Fig. 6, our method facilitates precise and effective 3D editing of specific objects within complex and cluttered scenes, a task that previous methods have found challenging. Our approach allows for the editing of objects of varying sizes and complexities, ranging from large items such as cabinets and furniture to smaller, more intricate objects like potted plants on tables. This versatility is achieved through our flexible camera pose generation strategy (*supp.* B.3.1), which allows for the generation of images suitable for diffusion.

Furthermore, rather than relying on 2D masks to identify and segment objects, we utilize open-set semantic queries to directly target them, resulting in more precise edits (*supp.* B.3.2). Experimental results demonstrate that our method can edit the target objects while preserving the integrity and consistency of the surrounding environment. This approach also highlights that constructing an open-set semantic field will be advantageous for the execution of downstream tasks.

Table 6: **A set of ablation studies** on Replica (Straub et al., 2019) (§3.5).

| | Dim | GPU Memory Usage | PSNR↑ | mIoU↑ | Size | mIoU ↑ | Pos-Sample | mIoU ↑ |
|---|---|---|---|---|---|---|---|---|
| *w/o* $\mathcal{F}$ | 3 | N GB | 32.88 | 72.56 | 50 | 87.34 | *w/o* $\mathcal{L}_{SCO}$ | 78.21 |
| | 6 | ∼(N + 12.0) GB | - | - | 100 | 93.62 | intra-frame | 85.96 |
| | 3 | (N + 1.5) GB | 32.94 | 74.29 | **200** | **95.14** | inter-frame | 87.83 |
| | 6 | (N + 1.7) GB | 33.07 | 87.95 | 500 | 94.96 | **All** | **95.14** |
| *w/* $\mathcal{F}$ | **16** | **(N + 2.4) GB** | **33.16** | **95.14** | 1000 | 95.38 | (c) Inter- and Intra-Frame | |
| | 32 | (N + 3.7) GB | 33.15 | 95.23 | (b) Pool Size | | Consistency Obj. | |

(a) Semantic Feature Dimension

## 3.5 ABLATION STUDY

Ablation studies are conducted on *Room 0* of Replica (Straub et al., 2019), following existing efforts.

**Key Component Analysis.** In Table 5, we validate the importance of our proposed components by attaching them one at a time. The 1st row reports the results of directly assigning a 3-dimensional feature to each Gaussian. Next, in the 2nd row, we utilize the semantic feature pool, resulting in complete experiment. Moreover, the 3rd row gives the results when applying intra-inter semantic consistency objective, and the objective leads to an improvement on semantic reconstruction. Finally, as shown in the 4th row, through semantic stability guidance, the biggest improvement is achieved, demonstrating the necessity of preventing from wrong features influence.

Table 5: **Quantitative Results of Key Component Analysis** (§3.5) on Replica (Straub et al., 2019). $\mathcal{F}$, $\mathcal{L}_{CO}$, $\mathcal{M}_{SG}$ represent semantic feature pool, intra-inter semantic consistency objective and semantic stability guidance, respectively.

| | PSNR ↑ | ATE RMSE ↓ | mIoU ↑ |
|---|---|---|---|
| *w/o* All | 32.88 | 0.30 | 72.56 |
| + $\mathcal{F}$ | 32.98 | 0.27 | 83.15 |
| + $\mathcal{L}_{CO}$ | 33.05 | 0.25 | 88.23 |
| + $\mathcal{M}_{SG}$ | **33.16** | **0.24** | **95.14** |

**Semantic Feature Dimension.** We examine how the semantic feature dimension affects semantic reconstruction, both with and without $\mathcal{F}$. As shown in Table 6a, without $\mathcal{F}$, using only 6-dimensional features leads to experimental failure. We report GPU memory usage across the framework, where N = 16.0 GB; while with $\mathcal{F}$, we include memory for $\mathcal{K}$ and $\mathcal{F}$. Notably, without $\mathcal{F}$, even 6-dimensional features consume more memory than those with it. Additionally, low-dimensional features are inadequate for high-quality reconstruction, while higher dimensions provide diminishing returns.

**Semantic Feature Pool Size.** Table 6b presents the effect of varying semantic feature pool sizes on semantic reconstruction quality. The results indicate that an overly small feature pool is insufficient to represent all the semantic features in the scene, while an overly large pool does not significantly improve the reconstruction quality. Therefore, the pool size of 200 that we used is sufficient to represent the entire scene.

**Intra- and Inter-Frame Semantic Consistency Objective.** We next study the impact of different intra- and inter-frame objective strategies. In Table 6c, results indicate that both strategies enhance semantic consistency. Notably, the inter-frame strategy achieves a greater improvement due to its ability to maintain continuity across frames.

## 4 CONCLUSION

We propose a novel method for generating a comprehensive metric-semantic 3D virtual world using multi-view stereo. By integrating 3D Gaussian representations with open-set semantics derived from 2D foundation models, our approach enables the creation of scalable and evolving 3D representations within a SLAM framework. To address the significant memory and computational challenges, we introduce Semantic Feature Aggregation. Furthermore, we incorporate the Intra-Inter Semantic Consistency Objective and Semantic Stability Guidance to ensure that the semantic reconstruction is both consistent and stable. Our experimental results demonstrate the potential of our open-ended 3D metric-semantic representation, opening up new possibilities for a wide range of downstream applications. We believe that this work marks a significant step forward in detailed, semantically rich 3D environments and efficient, expressive virtual world modeling.

REPRODUCIBILITY STATEMENT

We provide a comprehensive explanation of our method, covering Semantic Feature Aggregation, Intra-Inter Semantic Consistency Objective, and Semantic Stability Guidance (§2). Details on 3D scene editing are included in (*supp.* B), and we outline our implementation details in §3.1 and §C.1. The licenses for the assets used are reported in (*supp.* D). We promise code and instructions shall be made publicly available right after acceptance.

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

## SUMMARY OF THE APPENDIX

This appendix contains additional details for the ICLR 2025 submission, titled *Open-Ended 3D Metric-Semantic Representation Learning via Semantic-Embedded Gaussian Splatting*. The appendix is organized as follows:

- §A introduces related works to our framework.
- §B provies more details on our method.
- §C presents more experiment results, especially on real-world datasets.
- §D offers the licenses of assets we use.

## A  RELATED WORK

**3D Scene Representation.** In recent years, NeRFs (Mildenhall et al., 2020) have emerged as a pivotal advancement in the fields of 3D reconstruction and open-set semantic segmentation (Kerr et al., 2023; Kim et al., 2024; Liu et al., 2023; Engelmann et al., 2024), celebrated for their ability to synthesize high-quality novel views. However, despite these advancements, NeRF-based methodologies are inherently constrained by certain limitations. Training times remain relatively slow, and for open-set semantic segmentation, their implicit representations complicate precise 3D region identification. In contrast, 3D Gaussian Splatting (Kerbl et al., 2023), an explicit representation technique, offers a more suitable framework for accurate region identification, while maintaining reconstruction quality and significantly improving efficiency. Although several concurrent works (Shi et al., 2024; Qin et al., 2024; Zuo et al., 2024; Zhou et al., 2024; Qiu et al., 2024; Qu et al., 2024; Dou et al., 2024; Guo et al., 2024; Liao et al., 2024; Wu et al., 2024) have advanced this field, these methods face challenges when applied to arbitrary scenarios, particularly in large-scale environments or tasks requiring continuous scene expansion. In response, our approach leverages SLAM to achieve the reconstruction of any scene, aiming to achieve fast, efficient, and highly adaptable 3D reconstruction capabilities, thereby meeting the demands of a wider range of practical applications.

**Dense Visual SLAM.** A range of works (Sucar et al., 2021; Zhu et al., 2022; Rosinol et al., 2023; Yang et al., 2022; Zhang et al., 2023; Wang et al., 2023; Kong et al., 2023; Johari et al., 2023; Sandström et al., 2023) have furthered the development of SLAM with NeRFs (Mildenhall et al., 2020) through innovations like hierarchical multi-feature grids, uncertainty estimation, and improved loss functions. Nonetheless, these methods continue to grapple with the limitations posed by implicit representations. The advent of 3D Gaussian Splatting (Kerbl et al., 2023) has spurred a variety of works (Keetha et al., 2024; Matsuki et al., 2024; Yugay et al., 2023; Yan et al., 2024; Huang et al., 2024; Deng et al., 2024; Li et al., 2024b; Sandström et al., 2024; Hu et al., 2024; Ha et al., 2024; Sun et al., 2024b; Peng et al., 2024) that have made substantial strides in SLAM, resulting in enhanced reconstruction quality and processing speed. However, these works focus primarily on improving the RGB map, while overlooking the fact that a map with semantic information is more essential for expanding the applications of SLAM in various downstream tasks. To address this, both NeRF-based (Haghighi et al., 2023; Li et al., 2023b; Zhu et al., 2024b) and 3DGS-based (Li et al., 2024c; Ji et al., 2024; Zhu et al., 2024a) approaches have attempted to integrate semantics into SLAM. However, most of these methods are trained in a highly supervised approach on closed-sets data. Although they enrich the scene information, they lack the flexibility to continuously adapt to any scene and more downstream tasks. Our work endeavors to introduce precise open-set semantic features into SLAM systems, thereby significantly enhancing the system's ability to comprehend arbitrary scenes and facilitating more effective downstream tasks.

**Contrastive Semantic Learning.** To obtain open-set semantic features, we need to process each frame using a pre-trained model. Although this ensures the simplicity of using only RGB-D images inputs, the pre-trained model cannot guarantee the consistency of semantic features of the same object across different frames. Therefore, we introduce the idea of contrastive learning. In 2D self-supervised representation learning, instance discrimination (Dosovitskiy et al., 2014) has achieved substantial progress as a pre-training task for visual representations. While notable transfer learning performance has been demonstrated for image classification (Chen et al., 2020a;b; He et al., 2020), instance discrimination treats entire images as holistic entities, overlooking the complex internal structures of natural images. To address this, research has shifted towards pixel-level (Liu et al., 2020; Wang et al., 2021; Xie et al., 2021b) and object-level (Hénaff et al., 2021; 2022;

Wei et al., 2021; Wen et al., 2022; Xie et al., 2021a) discrimination, leading to the enhancement on the intrinsic structure of images and better transfer performance on dense prediction tasks. In contrast to 2D, self-supervised representation learning in 3D is still emerging. Early works (Hassani & Haley, 2019; Sanghi, 2020; Sauder & Sievers, 2019; Wang & Solomon, 2019) focused on object-centric point cloud data (Chang et al., 2015), but these methods do not facilitate 3D scene understanding (Xie et al., 2020). More recent studies (Hou et al., 2021; Huang et al., 2021; Wu et al., 2023; Xie et al., 2020; Yang et al., 2024; Zhang et al., 2021; Zhu et al., 2023) have developed 3D self-supervised representation learning on scene-centric data (Dai et al., 2017), significantly improving performance across a range of 3D scene perception tasks. Inspired by these advancements in contrastive learning, we introduce the idea of contrastive learning into semantic reconstruction. By applying 2D pixel-level semantic discrimination, we can reconstruct more accurate 3D semantic fields, thereby enhancing scene reconstruction quality and improving downstream task performance.

## B  MORE METHODS

### B.1  PREPROCESS WITH SAM AND CLIP

We accurately generate object masks with a state-of-the-art image segmentation model SAM (Kirillov et al., 2023). We then extract pixel-aligned CLIP (Radford et al., 2021) features for each segmented object.

### B.2  FRAMEWORK DETAILS

**Auxiliary Semantic Head.** Given a 2D semantic feature map $F$ rendered from semantic features, we train an semantic head to perform a classification task. Specifically, for each pixel $p$ in $F$, we classify it into one of the annotated categories present in the scene, resulting in a semantic map $S(p)$:

$$S(p) = \text{classifier}(F(p)). \tag{15}$$

We implement the classification for the quantitative comparison mentioned in §3.3.

**Selected Mapping Frames and Keyframes.** Instead of performing mapping at every frame, we adopt a strategy of mapping only at selected frames. Specifically, since the optimization of Gaussians tends to prioritize the most recent input frames, this can result in the system forgetting optimization results from earlier frames. To mitigate this forgetting issue caused by excessive mapping, we only initiate mapping when a substantial number of new Gaussians need to be added to the scene in the current frame. Additionally, if mapping has not occurred for several consecutive frames, the current frame is designated as a mapping frame to ensure complete scene optimization.

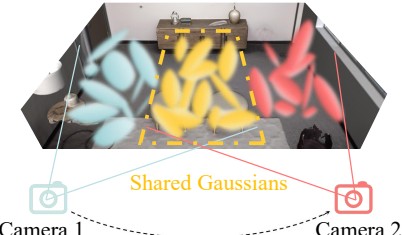

Figure 7: The forgetting problem during mapping.

We also maintain a set of keyframes, where every $n^{th}$ frame, along with the current frame, is stored as a keyframe. During mapping, we assess overlap by analyzing the point cloud of the current depth map and counting the number of points that fall within each keyframe's frustum to identify the keyframes most relevant to the current frame. For optimization, one of these highly relevant keyframes is randomly selected to optimize the current scene.

### B.3  DETAILS ON 3D EDITING

In previous research, the selective editing of individual objects within complex scenes has received limited attention. On one hand, diffusion-based editing methods often require that the input image clearly presents the object intended for editing. This poses challenges in images with multiple objects, where the diffusion process may struggle to accurately identify and isolate the desired object for modification (§B.3.1). On the other hand, there is a risk that diffusion might inadvertently alter parts of non-target objects, leading to unwanted changes in expansive open scenes (§B.3.2).

### B.3.1 CAMERA GENERATION

Given a center coordinate $o$ for an object, we can generate a sphere with a radius $r$, represented in spherical coordinates $(r, \theta, \varphi)$. Specifying an initial vector $\boldsymbol{v_0}$ and a given direction vector $\boldsymbol{v_1}$, we can obtain the rotation matrix between them with Rodrigues' rotation formula. Specifically, we have $\boldsymbol{v} = \boldsymbol{v_0} \times \boldsymbol{v_1}$, $c = \boldsymbol{v_0} \cdot \boldsymbol{v_1}$, and $s = \|\boldsymbol{v}\|$. Then we can get

$$R = I + K + K^2 \frac{1-c}{s^2}, \tag{16}$$

where

$$K = \begin{bmatrix} 0 & -\boldsymbol{v}_z & \boldsymbol{v}_y \\ \boldsymbol{v}_z & 0 & -\boldsymbol{v}_x \\ -\boldsymbol{v}_y & \boldsymbol{v}_x & 0 \end{bmatrix}. \tag{17}$$

Through this, we can transform any point on the surface of the sphere. We divide the radian values of the spherical coordinate system $\theta, \varphi$ evenly to generate camera positions on the sphere, and the transformation from spherical coordinates to Cartesian coordinates is achieved as follows:

$$x = r \sin\theta \cos\varphi, y = r \sin\theta \sin\varphi, z = r \cos\theta. \tag{18}$$

We then generate the camera pose matrix. With the positions of the cameras and the center coordinate $o$, we have the look direction vector **look_dir**, which indicates the direction the camera is pointing. Given an upper vector (such as $\boldsymbol{v_0}$) **up**, we derive the right vector as **right** = **up** × **look_dir**. To ensure the upper vector is completely orthogonal to both the look direction and right vector, we compute the cross product of the look direction and the right vector, yielding a corrected upper vector **up′** = **look_dir** × **right**, which ensures **right**, **up′**, **look_dir** form an orthonormal basis, representing the $x$, $y$, $z$ axes of the camera coordinate system respectively.

Then we can construct the view matrix:

$$View\_Matrix = \begin{bmatrix} \mathbf{right}_x & \mathbf{up}'_x & -\mathbf{look\_dir}_x & -\mathbf{right} \cdot \mathbf{camera\_pos} \\ \mathbf{right}_y & \mathbf{up}'_y & -\mathbf{look\_dir}_y & -\mathbf{up} \cdot \mathbf{camera\_pos} \\ \mathbf{right}_z & \mathbf{up}'_z & -\mathbf{look\_dir}_z & -\mathbf{look\_dir} \cdot \mathbf{camera\_pos} \\ 0 & 0 & 0 & 1 \end{bmatrix}. \tag{19}$$

After this, we utilize the Gaussian differentiable renderer to generate a sequence of cameras positioned around the target object. This allows us to render images centered on the object, meeting the requirements for diffusion-based editing.

### B.3.2 PRECISE EDITING

Editing specific objects without unintentionally affecting surrounding areas, such as the background, often involves using masks that restrict loss computation to the desired pixels. However, this approach faces significant challenges. **First**, updating masks during the editing process incurs substantial computational costs. Additionally, as the editing progresses, mask generation may become inaccurate, leading to cumulative errors. In contrast, static masks might not adequately support precise edits and could limit modifications that exceed the mask's boundaries. **Second**, within the rendering pipeline, the impact is not confined solely to the NeRF regions or Gaussians covered by the mask; parts not visible in the current view, which, however, theoretically lie inside the mask region, can still be inadvertently modified. This interaction complicates the management of error propagation, particularly when utilizing 2D masks.

To address these challenges, our methodology leverages the embedded semantic information in our Gaussian representation. This allows us to accurately identify the specific Gaussians that need modification. Specifically, we employ the CLIP model (Radford et al., 2021) to extract features from the input text, evaluate their similarity to the semantic features embedded in the Gaussians, and selectively refine the relevant Gaussians for the editing task. This approach effectively improves editing precision by directly interacting with the scene's underlying representation.

During the editing process, we randomly select a view obtained from §B.3.1 to render the original image $I_{ori}$, and utilize InstructPix2Pix (Brooks et al., 2023) to generate the edited image $I_{edit}$. We then compute the perceptual loss (Johnson et al., 2016) between them:

$$\mathcal{L}_{editing} = \text{Perceptual Loss}(I_{ori}, I_{edit}). \tag{20}$$

At each iteration, views are re-selected and modifications are applied to the previously edited scene. This iterative approach ensures a seamless editing outcome while preventing discrepancies across views that could arise from a single editing pass.

# C   MORE EXPERIMENT RESULTS

## C.1   EXPERIMENTAL SETUP

**Datasets.** The experiment results in this part are conducted on Replica (Straub et al., 2019), TUM-RGBD (Sturm et al., 2012) and ScanNet (Dai et al., 2017), with evaluations on 8, 5 and 6 scenes.

**Metrics, Baselines and Implementation Details.** We compare our method with the same baselines on the same metrics in §3.1. Implementation details are also the same, except that iterations of tracking and mapping are 100 and 30 for ScanNet, 200 and 30 for TUM.

## C.2   EXPERIMENTAL RESULTS ON SLAM

**Comparison of Camera Pose Estimation.** As shown in Table 8, our method markedly outshines existing NeRF-based approaches, even in the demanding scenarios characterized by the TUM-RGBD dataset, where sparse depth information and acute motion blur are prevalent. Despite the challenges similar to those encountered with TUM-RGBD, our approach still yields competitive results on ScanNet when assessed against the most advanced methods. The significant advancements underscored by our results, particularly in scenarios with low-quality inputs, show the effectiveness and potential of our method.

**Quantitative Analysis of Scene Reconstruction.** The rendering results on real-world data are presented in Table 9. The organized results reinforce the superior performance of our method over previous approaches, including Point-SLAM (Sandström et al., 2023), which demonstrated comparable results on the synthetic dataset. This robustly showcases the effectiveness of our method in real-world settings.

**Qualitative Comparison.** In Fig. 9 we show a comparison of our method with other methods NICE-SLAM (Zhu et al., 2022) and Point-SLAM (Sandström et al., 2023) on real-world datasets ScanNet (Dai et al., 2017) and TUM-RGBD (Sturm et al., 2012). This set of visual comparisons clearly demonstrates the enhancement in reconstruction quality achieved by our method, particularly since these results are obtained using real-world datasets. Our approach leads to more complete reconstructions of small objects and significantly reduces blurring and artifacts. The results strongly indicate the substantial potential of our method when applied in real-world scenarios.

**Other Qualitative Results.** Fig. 10 provides more results showing the color field and semantic field reconstruction. As seen, we are able to reconstruct color fields of high quality on Replica (Straub et al., 2019), which is also reflected in the high-quality geometric appearance. At the same time, we can reconstruct semantic fields with clear boundaries, which facilitates subsequent downstream tasks related to editing.

## C.3   MORE ABLATION STUDY

**Mapping Frame Selection.** We conduct an experiment without mapping frame selection. The results show that this strategy improves the quality of scene reconstruction, which is also reflected in the accuracy of the semantic field. Moreover, it aids in the estimation of camera poses.

Table 7: **Quantitative Results of More Ablation Study** (§C.3). MFS represents mapping frame selection.

|  | PSNR ↑ | ATE RMSE ↓ | mIoU ↑ |
|---|---|---|---|
| *w/o* MFS | 32.88 | 0.29 | 91.57 |
| **Ours** | **33.16** | **0.24** | **95.14** |

**Ablation Study on Semantic Features for 3D Editing.** The experiments are illustrated in Fig. 8. Without semantic guidance, editing attempts tend to affect the entire scene indiscriminately. This outcome is particularly problematic in large-scale environments, similar to those in our experiments. By integrating linguistic data, our technique gains the ability to selectively refine the designated objects while leaving the surrounding scene intact. This selective precision in editing highlights

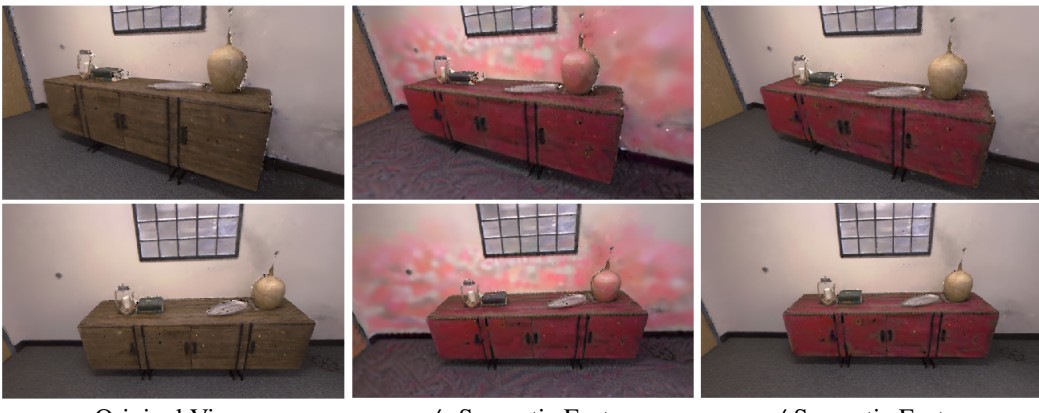

|  Original Views | *w/o* Semantic Features | *w/* Semantic Features |

Figure 8: **Ablation Study on Semantic Features for 3D Editing** (§C.3). Prompt: *Turn the cabinet red.*

the importance of semantic features in our methodology. The ablation study further demonstrates the accuracy of the reconstructed semantic information, validating its effectiveness in assisting with downstream tasks.

## D  ASSET LICENSE

We conduct our method on three indoor datasets (*e.g.*, Replica (Straub et al., 2019), TUM-RGBD (Sturm et al., 2012) and ScanNet (Dai et al., 2017)), and two pretrained models (*e.g.*, SAM (Kirillov et al., 2023) and CLIP (Radford et al., 2021)), which are all available for academic access. Replica (https://github.com/facebookresearch/Replica-Dataset) is released under this License. TUM-RGBD (https://cvg.cit.tum.de/data/datasets/rgbd-dataset) is released under this License. ScanNet (http://www.scan-net.org/) is released under this License. SAM (https://segment-anything.com/) is released under this License. CLIP (https://openai.com/index/clip/) is released under this MIT License.

Table 8: **Quantitative Results on Camera Pose Estimation** (§C.2) with baselines on TUM-RGBD (Sturm et al., 2012) and ScanNet (Dai et al., 2017) (ATE RMSE↓ [cm]).

| Methods | TUM-RGBD (Sturm et al., 2012) | | | | | | ScanNet (Dai et al., 2017) | | | | | | |
|---|---|---|---|---|---|---|---|---|---|---|---|---|---|
| | Avg. | fr1/ desk | fr1/ desk2 | fr1/ room | fr2/ xyz | fr3/ off. | Avg. | 0000 | 0059 | 0106 | 0169 | 0181 | 0207 |
| Vox-Fusion (Yang et al., 2022) | 11.31 | 3.52 | 6.00 | 19.53 | 1.49 | 26.01 | 26.90 | 68.84 | 24.18 | 8.41 | 27.28 | 23.30 | 9.41 |
| NICE-SLAM (Zhu et al., 2022) | 15.87 | 4.26 | 4.99 | 34.49 | 31.73 | 3.87 | 10.70 | 12.00 | 14.00 | 7.90 | **10.90** | 13.40 | **6.20** |
| Point-SLAM (Sandström et al., 2023) | 8.92 | 4.34 | **4.54** | 30.92 | 1.31 | **3.48** | 12.19 | **10.24** | **7.81** | 8.65 | 22.16 | 14.77 | 9.54 |
| SplaTAM (Keetha et al., 2024) | 5.48 | **3.35** | 6.54 | 11.13 | **1.24** | 5.16 | 11.88 | 12.83 | 10.10 | 17.72 | 12.08 | **11.10** | 7.46 |
| **Ours** | **5.25** | 3.29 | 5.86 | **10.95** | 1.29 | 4.87 | **10.23** | 14.56 | 9.20 | **7.82** | 11.59 | 11.11 | 7.12 |

Table 9: **Quantitative Comparison on Rendering Performance** (§C.2) with baselines on TUM-RGBD (Sturm et al., 2012) and ScanNet (Dai et al., 2017).

| Methods | Metrics | TUM-RGBD (Sturm et al., 2012) | | | | | | ScanNet (Dai et al., 2017) | | | | | | |
|---|---|---|---|---|---|---|---|---|---|---|---|---|---|---|
| | | Avg. | fr1/ desk | fr1/ desk2 | fr1/ room | fr2/ xyz | fr3/ off. | Avg. | 0000 | 0059 | 0106 | 0169 | 0181 | 0207 |
| Vox-Fusion (Yang et al., 2022) | PSNR↑ | 15.54 | 15.79 | 14.12 | 14.20 | 16.32 | 17.27 | 18.17 | 19.06 | 16.38 | **18.46** | 18.69 | 16.75 | 19.66 |
| | SSIM↑ | 0.632 | 0.647 | 0.568 | 0.566 | 0.706 | 0.677 | 0.673 | 0.662 | 0.615 | **0.753** | 0.650 | 0.666 | 0.696 |
| | LPIPS↓ | 0.502 | 0.523 | 0.541 | 0.559 | 0.433 | 0.456 | 0.504 | 0.515 | 0.528 | 0.439 | 0.513 | 0.532 | 0.500 |
| NICE-SLAM (Zhu et al., 2022) | PSNR↑ | 13.59 | 13.83 | 12.00 | 11.39 | 17.87 | 12.89 | 17.54 | 18.71 | 16.55 | 17.29 | 18.75 | 15.56 | 18.38 |
| | SSIM↑ | 0.545 | 0.569 | 0.514 | 0.373 | 0.718 | 0.554 | 0.621 | 0.641 | 0.605 | 0.646 | 0.629 | 0.562 | 0.646 |
| | LPIPS↓ | 0.494 | 0.482 | 0.520 | 0.629 | 0.344 | 0.498 | 0.548 | 0.561 | 0.534 | 0.510 | 0.534 | 0.602 | 0.552 |
| ESLAM (Johari et al., 2023) | PSNR↑ | 13.42 | 11.29 | 12.30 | 9.06 | 17.46 | 17.02 | 15.29 | 15.70 | 14.48 | 15.44 | 14.56 | 14.22 | 17.32 |
| | SSIM↑ | 0.599 | 0.666 | 0.634 | **0.929** | 0.310 | 0.457 | 0.658 | 0.687 | 0.632 | 0.656 | 0.696 | 0.653 | |
| | LPIPS↓ | 0.464 | 0.358 | 0.421 | **0.192** | 0.698 | 0.652 | 0.488 | 0.449 | 0.450 | 0.529 | 0.486 | 0.482 | 0.534 |
| Point-SLAM (Sandström et al., 2023) | PSNR↑ | 15.63 | 13.87 | 14.12 | 14.16 | 17.56 | 18.43 | 19.82 | 21.30 | 19.48 | 16.80 | 18.53 | 22.27 | 20.56 |
| | SSIM↑ | 0.665 | 0.627 | 0.592 | 0.645 | 0.708 | 0.754 | 0.751 | 0.806 | 0.765 | 0.676 | 0.686 | 0.823 | 0.750 |
| | LPIPS↓ | 0.538 | 0.544 | 0.568 | 0.546 | 0.585 | 0.448 | 0.514 | 0.485 | 0.499 | 0.544 | 0.542 | 0.471 | 0.544 |
| **Ours** | PSNR↑ | **21.76** | **22.43** | **19.88** | **20.26** | **24.77** | **21.48** | **20.13** | 21.15 | 19.27 | 18.46 | 21.44 | 18.83 | 21.63 |
| | SSIM↑ | **0.881** | **0.913** | **0.840** | 0.833 | **0.950** | **0.868** | **0.776** | 0.764 | 0.817 | 0.711 | 0.790 | 0.773 | 0.798 |
| | LPIPS↓ | **0.181** | **0.150** | **0.226** | 0.223 | **0.096** | **0.210** | **0.302** | 0.314 | 0.252 | 0.356 | 0.264 | 0.375 | 0.250 |

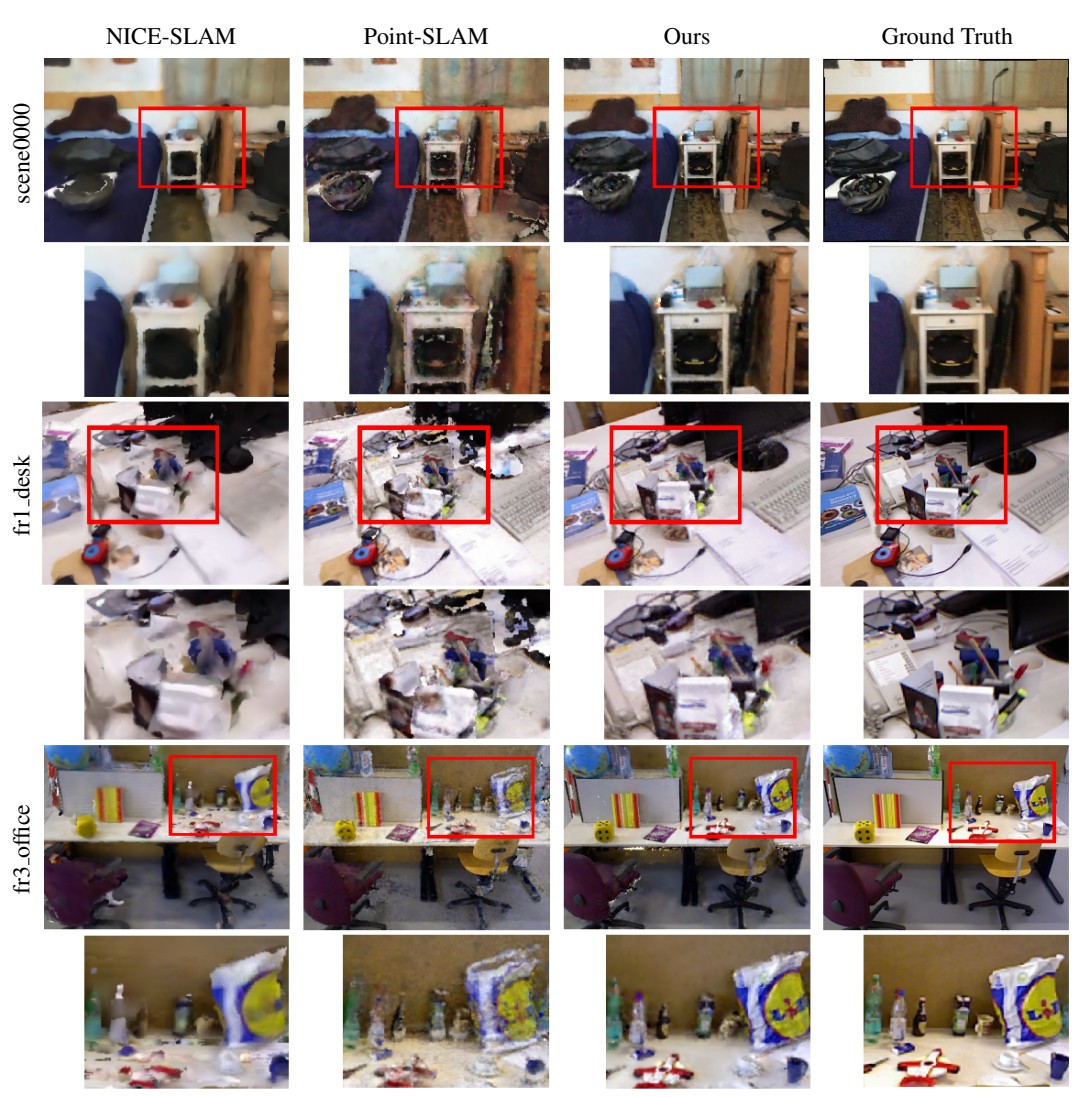

Figure 9: **Qualitative Comparisons of Rendering Performance** (§C.2) on ScanNet (Dai et al., 2017) and TUM-RGBD (Sturm et al., 2012).

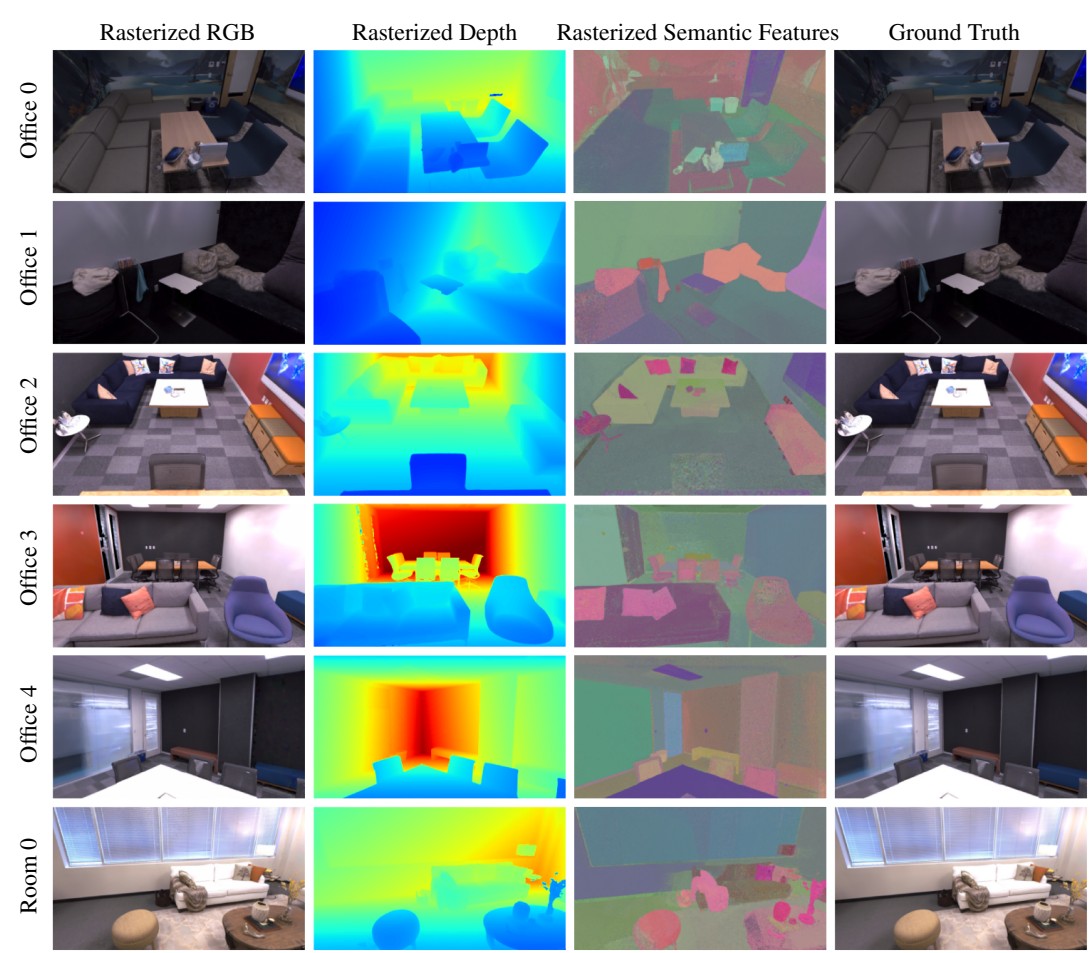

Figure 10: **Qualitative Results of Rendering Performance** (§C.2) of RGB, depth and semantic features on Replica (Straub et al., 2019).

