# OpenReview forum: "Open-Ended 3D Metric-Semantic Representation Learning via Semantic-Embedded Gaussian Splatting"
_ICLR.cc/2025/Conference — ICLR 2025 Conference Withdrawn Submission_

### Official Review · Reviewer_5Ngv · 2024-10-24

**Soundness:** 3
**Presentation:** 2
**Contribution:** 2
**Rating:** 3
**Confidence:** 5

**Summary:**

### Motivation
- Recent developments in SLAM and differentiable scene representations have converged into solutions that aims at capturing both the spatial and semantic layout of new scenes, based on visual observations.
- However, current methods still subordinate the semantic information (distilled frame-by-frame from large vision-language models) to the input visual modality, resulting in 3D representations that may _look_ accurate but suffer from semantic inconsistencies. Efficient solutions to reducing the high dimensionality of semantic embeddings (e.g., 512-dim vectors for CLIP) are also key to balance and efficiency.


### Claims
- The authors first introduce a codebook-based compression of language embeddings, i.e., making their model learn a fixed-size global pool of features, as well as low-dimensional key vectors mapping to the former.
- Then they extend existing 3DGS-based SLAM methods [Yan et al., 2024 ; Keetha et al., 2024 ; etc.] with an intra-/inter-frame semantic consistency supervision based on contrastive learning, i.e., enforcing the semantic features of Gaussians to be consistent across consecutive frame-projections (_intra-frame_) and consistent with neighboring projected features (_inter-frame_).
- They similarly verify the consistency of input language maps and use this consistency score to weigh the semantic loss (i.e., regions with low semantic consistency across consecutive frames are given less emphasis during training).



### Results
- Experiments on various datasets (Replica, TUM-RGBD, ScanNet) show that the propose method achieves SOTA performance in scene reconstruction, novel-view synthesis, and semantic segmentation.
- The authors also qualitatively demonstrate the applicability of their method to text-guided scene editing.
- Various ablation studies over the claimed contributions and their parameters are also provided, showing the statistical value of each component.

**Strengths:**

_(somewhat ordered from most to least important)_

### S1. Thorough Evaluation

The authors share results on 3 different datasets, using a set of metrics relevant to SLAM and semantic scene understanding (ATE/RMSE for tracking, PSNR/LPIPS/SSIM for image quality, and mIoU for semantic segmentation). They further share some convincing qualitative results on 3DGS scene editing, as well as various ablation studies w.r.t. their contributions and corresponding parameters.

### S2. Meaningful Iteration Over Existing 3D Semantic Consistency Schemes

3D/4D consistency of language features in Gaussian-splatting-based representations is a relatively new challenge, still under-explored. One of the proposed strategies (intra-/inter-frame contrastive learning) iterates over existing solutions (e.g., [Liao et al, 2024]), seemingly improving the overall robustness.

### S3. Good Reproducibility (minor)

The authors claim that their code will be released [L361].
Moreover, based on the implementation details that they provide, and leveraging open-sourced prior art as baseline (e.g., LangSplat or GOI), an expert in the art should be able to re-implement their work.

### S4. Well-Written & Illustrated Paper (minor)

The paper is overall well motivated and the methodology is easy to follow. Moreover, the provided diagrams clearly convey key points.

**Weaknesses:**

_(somewhat ordered from most to least important)_

### W1. No SOTA Review/Comparison w.r.t. Prior CLIP Dimensionality Reduction Methods

The first key contribution claimed by the authors relates to tackling the high dimensionality of CLIP language features for their integration into 3DGS representations. However, this topic has already been covered in various way by prior art. I.e., all existing language-embedded 3D Gaussian models have to deal one way or another with the prohibitively large size of CLIP features. Some solutions leverage the lower-dimensional latent space of scene-specific autoencoders [Qin et al., 2024], others apply PCA [Xu et al., 2024], etc., and some already implemented codebook-based strategies similar to the one proposed here [Qu et al., 2024; Shi et al., 2024, Liao et al., 2024].

The fact that the authors neither compare to these existing CLIP-dimension-reduction techniques, nor even mention them, is problematic, showing a lack of contextualization relative to prior art and questioning the actual scope of their technical contributions.

### W2. No SOTA Review/Comparison w.r.t. 3D Language Consistency/Stability Methods

The authors also failed to disclose prior art tackling 3D semantic consistency in language-embedded 3DGS representations. For example, CLIP-GS [Liao et al., 2024] seemingly applies a similar regularization of 3D language features over consecutive frames, as well as voting-based regularization of CLIP features over segmented regions, similar to the proposed "semantic stability guidance".
While the proposed contrastive-learning approach may be considered somewhat novel, it is also a relatively low-hanging extension of CLIP-GS' solution.

Similar to the previous point `W1`, the key issue here is the lack of contextualization/comparison done by the authors w.r.t. their technical claims. How different is their consistency/stability schemes from CLIP-GS ones? Which ones are actually better? — Here again, the readers are left in the dark.


### W3. SLAM Framework Heavily Based on Existing Solutions

While not claimed as a contribution per se, the authors similarly poorly contextualized their SLAM framework, with no references provided for the parts that they borrowed.

E.g., their tracking step—composed of a linear forward projection to initialize the camera parameters (Equation 7), followed by a multi-modal rendering-based optimization (Equation 8)—appears heavily based on prior work, e.g., SplaTAM [Keetha et al, 2024] and SGS-SLAM [Li et al, 2024c].


### W4. Minor Remarks

- The authors do not provide any insight on possible limitations / failure cases / societal impact.
- The wrapping of text around some large tables (Tables 2 and 3 especially) and the reduced margins makes some paragraphs hard to read.
- Equation 2: Symbols $\mu^{2D}$ and $r^{2D}$  are introduced there but never used elsewhere.
- Equations 3, 4, and 6: the 2D projection of $g_i(p)$ should be used, rather than the 3D position.
- $\mathcal{M}_{SG}$ could be formally defined for clarity/reproducibility (i.e., no equation is currently provided for this similarity function).


#### **Additional References:**

- Xu et al. "TIGER: Text-Instructed 3D Gaussian Retrieval and Coherent Editing." arXiv preprint arXiv:2405.14455 (2024).
- Liao et al. "CLIP-GS: CLIP-Informed Gaussian Splatting for Real-time and View-consistent 3D Semantic Understanding." arXiv preprint arXiv:2404.14249 (2024).

**Questions:**

_see **Weaknesses** for key questions/remarks._


### Q1. Pool Size

How is the pool size $M$ decided? Some insight on how to fix this hyperparameter for new scenes would be helpful to readers.

### Q2. Similarity Metrics

Which similarity metric is used for simVec$(\cdot, \cdot)$? Is it cosine similarity similar to other codebook methods?

---

### Official Review · Reviewer_XZUm · 2024-10-27

**Soundness:** 2
**Presentation:** 2
**Contribution:** 2
**Rating:** 3
**Confidence:** 5

**Summary:**

This paper introduces an open-ended metric-semantic representation learning framework based on 3D Gaussian Splatting. Authors propose to learn semantics by aggregating from a condensed, fixed-sized semantic pool and apply semantic consistency through contrastive learning.

**Strengths:**

1.This paper presents an open-ended 3D metric-semantic SLAM based on 3D Gaussian representation.
2.Experimental comparisons with other methods and ablation studies are conducted on three datasets.

**Weaknesses:**

1.The paper claims that the proposed SLAM method outperforms existing NeRF-based and 3DGS-based approaches. However, in camera pose estimation, the accuracy of CG-SLAM [ECCV'24] and RTG-SLAM [SIGGRAPH'24] is much higher than your method. Their average accuracy is 0.27 and 0.18 respectively, while the accuracy of this paper is 0.30 in Replica dataset (Table 2). For reconstruction performance, SGS-SLAM [ECCV'24], which is also a semantic SLAM system based on 3D Gaussian Splatting, has higher reconstruction accuracy than this paper. For rendering performance, other Gaussian SLAM methods, such as Gaussian-SLAM and MonoGS[CVPR'24], outperform your method. The results of this paper fail to prove the effectiveness of your method.
2.In SLAM, real-time performance is very important. This paper does not show the running speed of the system.
3.What’s the memory usage of storing the entire scene?
4.This paper does not compare visualization results with existing Gaussian SLAM methods, such as SplaTAM [CVPR'24] and MonoGS [CVPR'24], which are all open source.
5.In terms of innovation, the concept of contrastive learning has been widely used in semantic Gaussians, and the contrastive learning innovation in this paper is just constructing a loss. The innovation of the paper is relatively poor.

**Questions:**

1.The results of this paper fail to prove the effectiveness of your method. In camera pose estimation, the accuracy of CG-SLAM [ECCV'24] and RTG-SLAM [SIGGRAPH'24] is much higher than your method. For reconstruction performance, SGS-SLAM [ECCV'24], which is also a semantic SLAM system based on 3D Gaussian Splatting, has higher reconstruction accuracy than this paper. For rendering performance, other Gaussian SLAM methods, such as Gaussian-SLAM and MonoGS[CVPR'24], outperform your method.
2.What’s the running speed of the system?
3.What’s the memory usage of storing the entire scene?
4.This paper does not compare visualization results with existing Gaussian SLAM methods, such as SplaTAM [CVPR'24] and MonoGS [CVPR'24].
5.In terms of innovation, the concept of contrastive learning has been widely used in semantic Gaussians, and the contrastive learning innovation in this paper is just constructing a loss. The innovation of the paper is relatively poor.

---

### Official Review · Reviewer_zEbS · 2024-11-03

**Soundness:** 3
**Presentation:** 3
**Contribution:** 3
**Rating:** 8
**Confidence:** 4

**Summary:**

This paper describes a method for generating a metric-semantic 3D virtual world using multi-view stereo. By integrating 3D Gaussian representations with open-set semantics derived from 2D foundation models, it enables the creation of scalable and evolving 3D representations using SLAM. To address the significant memory and computational challenges, it introduces Semantic Feature Aggregation and incorporates the Intra-Inter Semantic Consistency Objective and Semantic Stability Guidance to ensure that the semantic reconstruction
is both consistent and stable. The proposed open-ended 3D metric-semantic representation offers new possibilities for a wide range of downstream applications and expressive virtual world modeling.

**Strengths:**

1. This paper presents a 3D GS-based open-ended 3D metric-semantic representation learning framework, which are optimized within a SLAM approach. The proposed method mainly focus on 2 main challenges: (1) improving the efficiency of semantic representation for entire scenes, and (2) resolving inter-frame inconsistencies across different segmented views.
2. The paper is well-structured and clearly presented, with precise language and notation throughout each section. The inputs, outputs, and technical details of each proposed module are described in a manner that is easy to follow and understand.
3. Linking each Gaussian primitive with a low-dimensional key effectively reduces memory consumption. Additionally, only features from distinct categories are stored in the feature pool, indicating that objects within the same category share common semantics. This design is efficient, well-structured, and avoids redundancy.
4. The “Intra-Inter Semantic Consistency Objective” module effectively leverages multi-view observations to ensure continuity and consistency within the learned semantic field. Meanwhile, this enhanced semantic field can also contribute to the optimization of mapping.
5. In the experimental section, the authors provide comprehensive quantitative results, as well as qualitative results across various downstream tasks. This thorough evaluation highlights the method’s potential for a wide range of applications, including 3D scene editing and other related tasks.

**Weaknesses:**

1. In line 231, How to determine the "well-reconstructed parts of the map"? It would be good to provide more detailed explanations? One possible way I think, as suggested in SplaTam based on rendering errors of different image parts.
2. In the “Intra-Inter Semantic Consistency Objective” module, the underlying insight for intra-frame consistency is that pixels belonging to the same object should share identical semantic features. However, the method appears to assume that the segmentation of each frame is consistently accurate. In practice, pixels on the edges of a 2D mask often exhibit high uncertainty, which may require more careful processing to ensure robustness.

**Questions:**

See above.

---

### Official Review · Reviewer_d1Yy · 2024-11-03

**Soundness:** 3
**Presentation:** 3
**Contribution:** 3
**Rating:** 5
**Confidence:** 5

**Summary:**

The paper presents a framework for open-ended semantic 3D representation learning using 3D Gaussians, optimized within a SLAM system. The proposed approach distills open-set semantics from 2D models, like CLIP and SAM, into a scalable 3D Gaussian representation. By introducing a fixed-size semantic feature pool and employing contrastive learning for consistency, the model reduces memory usage and enhances semantic coherence. Experimental results show this framework’s capability to perform accurate scene rendering, object tracking, and 3D scene editing, representing a step forward in creating detailed 3D virtual worlds.

**Strengths:**

- The framework proposes to use Gaussian Splatting and feature distillation within SLAM to solve the problem of open-vocabulary segmentation/editing in 3D in real-time, which addresses a key challenge in the field.

- The paper proposes the use of a fixed-size semantic pool with 3D Gaussians, incorporating regularization for Intra-Inter Semantic Consistency and guidance for Semantic Stability to achieve better accuracy.

- The paper conducts comprehensive experiments with promising results, demonstrating the framework's utility in applications such as object tracking and 3D scene editing, highlighting its potential for practical use.

**Weaknesses:**

- Although this application is new, it is not novel to apply distillation from a 2D-trained model into 3D, and it seems this paper addresses a similar problem in feature distillation as seen in methods such as LangSplat and Feature-3DGS.

- I have significant concerns about the writing: Section 2.3 should be moved to the preliminary part. Figure 1 and Figure 7 does not convey much new information compared to earlier literature.

- The experiments lack clarity: it is unclear why feature distillation improves both tracking and rendering accuracy, as SLAM uses a dense RGB-D sequence, which is sufficient for mapping, and there is no new method for the interaction between mapping and features (Tab. 5). Additionally, CLIP only predicts image-level embeddings, so how is it possible to integrate the CLIP model into 3D-GS directly (e.g., pixel-aligned CLIP in line 883)?

- It is crucial to upload a supplementary video to visualize the effects of distillation consistency and the quality of 3D segmentation and related tasks, whereas they are missing.

**Questions:**

Please consider the questions posed under weaknesses regarding the novelty claim, the writing issues, and the need for clearer explanations of the experiments.

---

### Note · Authors · 2024-11-15

I have read and agree with the venue's withdrawal policy on behalf of myself and my co-authors.